# GRAFTING VISION TRANSFORMERS

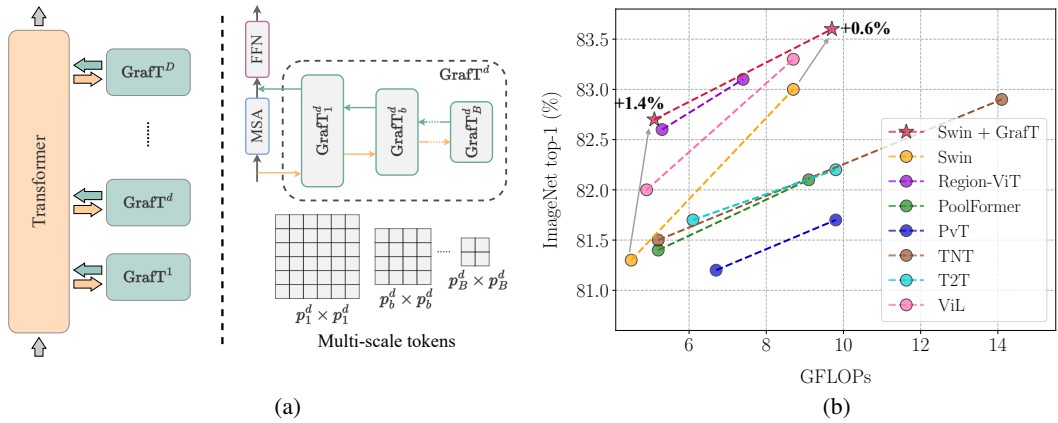

Figure 1: We introduce GrafT, an add-on component which makes use of global and multi-scale dependencies at arbitrary depths of a network. (a) An overview of how GrafT modules are branched-out (or grafted) from a backbone Transformer. Each GrafT may consider multiple scales of features (i.e., token representations), widening a network efficiently while relying on the backbone to perform most of the computations. It can be adopted to both homogeneous (*e.g.,* ViT (Dosovitskiy et al., 2020)) and pyramid (*e.g.,* Swin (Liu et al., 2021)) architectures. (b) Performance-complexity trade-off of our Swin+GrafT, in comparison with previous related methods. GrafT shows a considerable performance gains with a minimal increment in complexity.

## ABSTRACT

Vision Transformers (ViTs) have recently become the state-of-the-art across many computer vision tasks. In contrast to convolutional networks (CNNs), ViTs enable global information sharing even within shallow layers of a network, i.e., among high-resolution features. However, this perk was later overlooked with the success of pyramid architectures such as Swin Transformer, which show better performance-complexity trade-offs. In this paper, we present a simple and efficient add-on component (termed **GrafT**) that considers global dependencies and multi-scale information throughout the network, in both high- and low-resolution features alike. GrafT can be easily adopted in both homogeneous and pyramid Transformers while showing consistent gains. It has the flexibility of branching-out at arbitrary depths, widening a network with multiple scales. This grafting operation enables us to share most of the parameters and computations of the backbone, adding only minimal complexity, but with a higher yield. In fact, the process of progressively compounding multi-scale receptive fields in GrafT enables communications between local regions. We show the benefits of the proposed method on multiple benchmarks, including image classification (ImageNet-1K), semantic segmentation (ADE20K), object detection and instance segmentation (COCO2017). Our code and models will be made available.

## 1 INTRODUCTION

Self-attention mechanism in Transformers (Bello et al., 2019) has been widely-adopted in language domain for some time now. It can look into pairwise correlations between input sequences, learning long-range dependencies. More recently, following the seminal work in Vision Transformers (ViT) (Dosovitskiy et al., 2020), the vision community has also started exploiting this property,

showing state-of-the-art results on various tasks including classification, segmentation and detection, outperforming convolutional networks (CNNs) (He et al., 2016; Tan & Le, 2019; Howard et al., 2017; Sandler et al., 2018; Howard et al., 2019; Brock et al., 2021). Motivated by this success, many variants of vision transformers (*e.g.,* DeiT (Touvron et al., 2021), CrossViT (Chen et al., 2021), TNT (Han et al., 2021)) emerged, inheriting the same homogeneous structure of ViT (i.e., a structure w/o downsampling). However, due to the quadratic complexity of attention, such a structure becomes expensive, especially for high-resolution inputs and does not benefit from the semantically-rich information present in multi-scale representations.

To address these shortcomings, Transformers with pyramid structures (i.e., structures w/ downsampling) such as Swin (Liu et al., 2021) were introduced with hierarchical downsampling and window-based attention, which can learn multi-scale representations at a computational complexity linear with input resolution. As a result, pyramid structures become more suited for tasks such as segmentation and detection. However, still, multiple scales arise deep in to the network due to stage-wise downsampling, meaning that only the latter stages of the model may benefit from them. Thus, we poise the question: what if we can introduce multi-scale information even at early stages of a Transformer, without incurring a heavy computational burden?

Previous work has also looked into the direction above, both in CNNs (Szegedy et al., 2015a) and in Transformers (Chen et al., 2021; 2022). However, models such as CrossViT requires carefully tuning the spatial ratio of two feature maps in two branches and RegionViT needs considerable modifications to handle multi-scale training. To mitigate these issues, in this paper, we propose a simple and efficient add-on component called **GrafT** (see Figure 1-(a)). It can be easily adopted in existing homogeneous or pyramid architectures, enabling multi-scale features throughout a network (even in shallow layers) and showing consistent performance gains, while being computationally lightweight. GrafT is applicable at any arbitrary layer of a network. It consists of three main components: (1) a *left-right pathway* for downsampling, (2) a *right-left pathway* for upsampling, and (3) a *bottom-up connection* for information sharing at each scale. The left-right pathway uses a series of average pooling operations to create a set of multi-scale representations. For instance, if GrafT is attached to a layer with $(56 \times 56)$ resolution, it can create scales of $(28 \times 28)$, $(14 \times 14)$ and $(7 \times 7)$. We then process information at each scale with a *L-MSA* block, a local self-attention mechanism (*e.g.,* window-attention)— which becomes global-attention in the coarsest scale, as window-size becomes the same as the resolution. Next, the right-left pathway uses a series of learnable and window-based bi-linear interpolation (*W-Bilinear*) operations to generate high-resolution features by upsampling the low-resolution outputs of L-MSA— which contains global (or high-level) semantics extracted efficiently, at a lower resolution. Such upsampled features are merged with high-resolution features of the branch-to-the-left, which contain lower-level semantics, as also done in Feature Pyramid Networks (Lin et al., 2017). Refer to Figure 2-(b) for a detailed view.

GrafT is unique in the sense that it can extract multi-scale information at any given layer of a Transformer, while also being efficient. It relies on the backbone to do the heavy-lifting, by using a minimal computation overhead within grafted branches, in contrast to having completely-separate branches as in CrossViT (Chen et al., 2021). In our evaluations, we show the benefits of GrafT in both homogeneous (ViT) and pyramid (Swin) architectures, across multiple benchmarks: on ImageNet-1K (Deng et al., 2009), **+3.9**% with ViT-T+GrafT, **+1.4**% with Swin-T+GrafT, and **+0.5**% with , on COCO2017 (Lin et al., 2014), **+1.1** AP[b] for object detection and **+0.8** AP[m] for instance segmentation with Swin-T+GrafT, and on ADE20K (Zhou et al., 2017) semantic segmentation, **+1.0** mIOU[ss], **+1.3** mIOU[ms] with Swin-T+GrafT. Figure 1-(b) shows the performance-complexity trade-off Swin-T+GrafT on ImageNet-1K.

## 2 GRAFTING VISION TRANSFORMERS

Our goal is to provide multi-scale global information to the backbone transformer from the bottom layer so that high-level semantics from GrafT can help the transformer to construct more efficient features. Since Graft is modular, it can be applied to various transformer architectures. We select two representative transformers, ViT (Dosovitskiy et al., 2020) based on a homogeneous structure and Swin (Liu et al., 2021) based on a pyramid structure, to show that GrafT is a general purpose module.

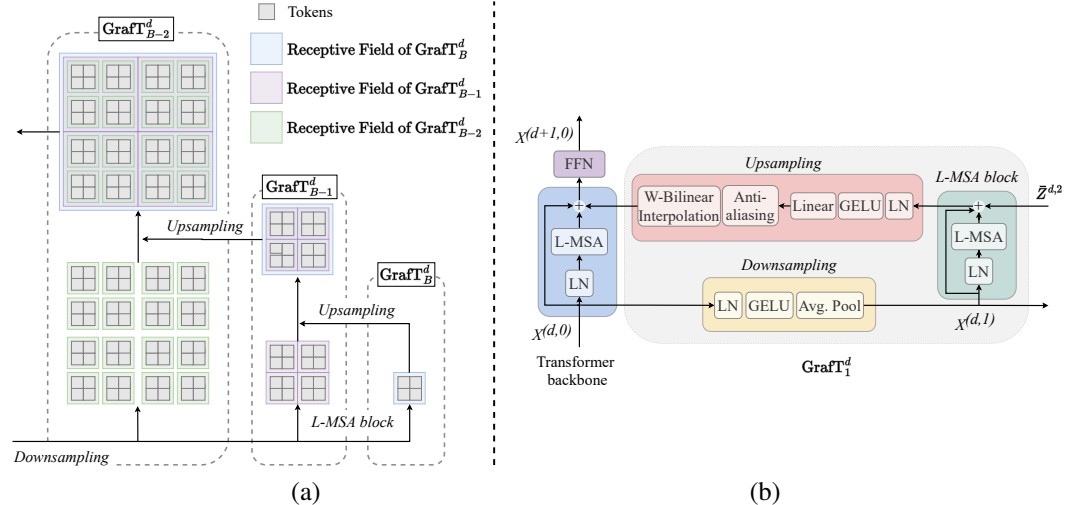

Figure 2: Detailed view of GrafT. **(a):** Here, we show the receptive field of attention mechanisms in each scale and how it changes by merging with information from the corresponding lower-resolution scale (i.e., from branch-to-the-right). Multiple scale are created with downsampling, processed with local attention (in *L-MSA*)— which becomes global attention in coarsest scale (GrafT$_B^d$), and merged back with upsampling. When a lower-resolution representation is upsampled and merged, the effective receptive field increases, essentially giving access to efficiently-extracted global (or larger-local) information. **(b):** We present all the components and grafting/ merging points in GrafT. We graft prior to self-attention block in the backbone, and merge prior to FFN so that we can reuse the heavy computations. Downsampling (*left-right pathway*) uses light-weight average pooling to create a lower-resolution features, whereas upsampling (*right-left pathway*) uses learnable window-based bi-linear interpolation (*W-Bilinear*) to upscale. The processing unit within a GrafT module is a *L-MSA* block, which performs local-attention. When merging features to higher-resolution, we use element-wise addition.

## 2.1 OVERALL ARCHITECTURE

The overall architecture of GrafT is illustrated in Figure 2-(a). We start with a backbone transformer (*i.e.,* ViT or Swin) and simply attach GrafT to some vertical layers of the backbone transformer. For an input image with size of $H \times W \times 3$, the patch tokens to the first vertical layer is $\frac{H}{4} \times \frac{W}{4} \times C$ after the patch embedding.

GrafT is a horizontal pyramid structure which consists of left-right pathway (series of GrafT downsampling), right-left pathway (series of GrafT upsampling), and bottom-up connection (multiple GrafT L-MSA) as shown in Figure 2-(b). The input feature to GrafT goes through left-right pathway (downsampling), right-left pathway (upsampling), bottom-up connection (L-MSA) and becomes a feature having strong high-level semantics at multiple scales which then gets fused into the backbone transformer. Specifically, for the GrafT attached to the vertical layer at $S^{th}$ stage, the input feature to the GrafT has the size of $\frac{H}{4r^{S-1}} \times \frac{W}{4r^{S-1}} \times C$ where $r = 2$ for pyramid structure and $r = 1$ for homogeneous structure. Then, we fuse the feature from GrafT to the original backbone, which will be described in the later section.

## 2.2 TRANSFORMER+GRAFT BLOCKS

In this section, we describe each operation in Figure 2-(b). Let $X^{d,b}$ as an input tensor at vertical layer $d$ and horizontal downsampling level $b$ in GrafT with the shape: $X^{d,b} \in \mathbb{R}^{H^b \times W^b \times C}$. $H^b$, $W^b$, $C$ is the height, width, channels of the feature map $X^{d,b}$ at a horizontal level $b$ respectively.

**Left-right pathway (downsampling):** The left-right pathway creates serial feature maps at several scales with the downsampling rate r which follows the vertical pyramid downsampling rate. For example, in the first stage of Swin+GrafT, GrafT creates three downsampled feature maps $\{X^1, X^2, X^3\}$ by downsampling the input feature map $X^0$ from the backbone with downsampling

rate 2. In left-right pathway, we use adaptive average pooling for downsampling:

$$X^{d,b+1} = A(X^{d,b}) = \rho(\text{GELU}(\text{LN}(X^{d,b}))) \tag{1}$$

$A$ is a lightweight downsampling function which consists of LayerNorm ($LN$), $GELU$, and adaptive average pooling ($\rho$), which is more cost-efficient than other downsampling methods such as cross attention and linear projection (see Table 5). It maps the input $X^{d,b}$ to downsampled feature $X^{d,b+1}$: $\mathbb{R}^{H^b \times W^b} \mapsto \mathbb{R}^{H^{b+1} \times W^{b+1}}$ where $H^{b+1} < H^b, W^{b+1} < W^b$. $X^{b+1}$ is the input at the horizontal downsampling level $b+1$. Downsampling happens sequentially over horizontal layers to progressively abstract the fine information in coarse features.

**Right-left pathway (upsampling):** The right-left pathway hallucinate serial higher resolution feature maps by upsampling low resolution feature maps. These upsampled feature maps are enhanced by feature maps from bottom-up connection that retain spatially more accurate activations and lower-level semantics. This enhancement process is similar to FPN (Lin et al., 2017). In right-left pathway, GrafT upsampling uses learnable W-Bilinear (window-base bilinear) interpolation which is more cost-efficient than other upsampling methods such as cross attention and nearest neighbor interpolation as shown in Table 5. Learnable W-Bilinear interpolation solves the aliasing problem by embedding anti-aliasing weights, the sigmoid of positional embeddings, in the feature maps. Anti-aliasing weights learns perturbations for each grid in the feature map that can prevent aliasing effect. W-Bilinear interpolation is adopted to address the semantics discontinuity between local regions. Finally, our upsampling is defined as:

$$\bar{Z}^{d,b+1} = \Phi(\tilde{Z}^{d,b+1}) = \Phi(E_{aa} \odot \alpha(Z^{d,b+1})) = T(Z^{d,b+1}), \quad \bar{Z}^{d,b+1}_{u_m,v_n} = \Phi(\tilde{Z}^{d,b+1}_{i_m,j_n}) \tag{2}$$

$T$ is a lightweight upsampling function mapping the representation back to the spatial resolution in the previous horizontal level. This function is designed to upsample output features from L-MSA. First, given the tensor $Z^{d,b+1}$, the output from L-MSA, channel mixing ($\alpha$) is applied to align channels before interpolation. $\alpha$ consists of LayerNorm ($LN$), $GELU$, and a linear layer. Next, anti-aliasing embeddings ($E_{aa}$) are multiplied to resolve the aliasing problem. The proposed anti-aliasing embeddings are the output of sigmoid function on position embeddings $Z^{j+1}_{pos} \in R^{H^{j+1} \times W^{j+1} \times C}$. It learns to provide perturbations in the spatial dimension that prevents interpolation from suffering the aliasing problem. It is a simpler and lighter method compared to 3x3 convolutions (Lin et al., 2017). Lastly, W-Bilinear interpolation ($\Phi$) maps $\tilde{Z}^{d,b+1} \in \mathbb{R}^{H^{b+1} \times W^{b+1}}$ to $\bar{Z}^{d,b+1} \in \mathbb{R}^{H^b \times W^b}$. It can be described as mapping the low resolution feature in each $(m, n)$th window $\tilde{Z}^{d,b+1}_{i_m,j_n}$ into high resolution feature in $(m, n)$th window $\bar{Z}^{d,b+1}_{u_m,v_n}$. $(i_m, j_n)$ is a spatial position $(i, j)$ in $(m, n)$th window where $i \in I, j \in J, m \in M, n \in N$. $(u_m, v_n)$ is a spatial position $(u, v)$ in $(m, n)$th window where $u \in r_h I, v \in r_w J$. $r_h, r_w$ are the height and width ratio between feature resolutions in two consecutive horizontal levels. Upsampling happens sequentially over horizontal layers to progressively upscale spatial resolution of coarse features.

**L-MSA in GrafT:** The local self-attention method from the backbone transformer, limits the self-attention in a local region which causes the discrepancy of semantics at the boundary of local regions. Therefore, bilinear interpolation is applied in each local region multiple times instead of the entire feature map at once. The bottom-up connection merges lower-level feature maps enhanced by L-MSA and higher-level feature maps upsampled by W-Bilinear interpolation by element-wise addition. The merging process iterates until it generates the high-level feature map that has the same spatial size as the input feature map $X^0$ from the backbone. In the example above, the coarsest feature map $X^3$ goes through L-MSA and becomes $\bar{Z}^3$, a feature map having the highest-level semantics. $\bar{Z}^2$ is generated by merging upsampled $\bar{Z}^3$ with output feature from L-MSA which applies local self-attention on $X^2$. This process iterates until the finest feature map $\bar{Z}^0$ which has the same spatial size of the $X^0$ is produced. $\bar{Z}^0$ is then fused into the output feature map from L-MSA in the backbone transformer and proceed to FFN block.

$$Z^{d,b+1} = X^{d,b+1} + [\text{L-MSA}(\text{LN}(X^{d,b+1})) + \bar{Z}^{d,b+2}] \tag{3}$$

It is a simple and light block that fuses multi-scale features. It uses a standard window-based MSA from Swin (Liu et al., 2021) to encode fine features $X^{d,b+1}$. It uses simple element-wise addition to

fuse this fine feature and coarse feature $\bar{Z}^{d,b+2} \in \mathbb{R}^{H^{b+1} \times W^{b+1}}$ coming from one deeper horizontal level. Lastly, Skip connection is added to produce output $Z^{d,b+1} \in \mathbb{R}^{H^{b+1} \times W^{b+1}}$

**Bottom-up connection with multiple high-level semantics:** In GrafT, L-MSA uses local self-attention to enhance downsampled feature maps but it limits receptive field within each local region. It is important to exchange information among local regions so that feature can embed not only local structure but also global structure. In the left side of Figure 2, the features at the bottom are the output of GrafT L-MSA where receptive field is limited to each local region. The feature map at $b+2$ level $\tilde{Z}^{b+2}$ has a receptive field, a blue color line, that covers the entire feature map. The feature map at $b+1$ level, $\tilde{Z}^{b+1}$, originally suffers the information discrepancy between local regions due to local receptive field. When $\tilde{Z}^{b+2}$ is merged into $\tilde{Z}^{b+1}$, $\tilde{Z}^{b+1}$ inherits a receptive field from $\tilde{Z}^{b+2}$ and this newly added global receptive field provides higher-level semantics that can understand the relation between local regions. $\tilde{Z}^{b}$, the feature map at $b$ level, also originally suffers the information discrepancy between local regions. When $\tilde{Z}^{b+1}$ is merged into $\tilde{Z}^{b}$, $\tilde{Z}^{b}$ inherits receptive fields from both $\tilde{Z}^{b+1}$ $\tilde{Z}^{b+2}$ and this newly added receptive fields provides multi-scale high-level semantics that can understand the relation between local regions in multiple aspects. Multi-scale receptive fields which are progressively generated from multi-scale features resolve the drawback of local receptive field formed by L-MSA in the backbone transformer. The L-MSA in the original backbone transformer is defined as:

$$Y^{d,0} = X^{d,0} + [\text{L-MSA}(X^{d,0}) + \bar{Z}^{d,1}] \tag{4}$$

L-MSA is the local multi-self attention that the transformer in the main branch is using to encode fine feature $X^{d,0} \in \mathbb{R}^{H^0 \times W^0 \times C}$. The encoded fine feature is element-wise added with coarse feature $\bar{Z}^{d,1}$ from the GrafT. Lastly, skip connection is added to produce output $Y^{d,0} \in \mathbb{R}^{H^0 \times W^0 \times C}$. It is interesting that a simple element-wise addition successfully fuse the fine feature in the main branch and the coarse feature from GrafT. Thanks to the power of GrafT upsampling method for the robust fusion of the multi-scale features encoded by various MSA. Then, we use the same FFN in the backbone transfomers:

$$X^{d+1,0} = Y^{d,0} + \text{MLP}(\text{LN}((Y^{d,0}))) \tag{5}$$

It is a standard transformer block performing channel mixing via LayerNorm ($LN$) and $MLP$ on the output of L-MSA block and adds the skip connection to generate output $X^{d+1,0}$ which is the input to the next vertical layer.

**Computation complexity:** Average pooling and bilinear interpolation runs in $\Theta(HW)$ and L-MSA in GrafT follows the complexity of L-MSA in the backbone transformer. Since L-MSA is more complex than $\Theta(HW)$, the complexity of transformer+GrafT is equal to the complexity of the pure backbone transformer. For example,

$$\Omega(\text{Swin}) = \Omega(\text{Swin+GrafT}) = 12HWC^2 + 2M^2HWC \tag{6}$$

where H,W is the width and height of feature map and M is the size of window.

## 3 EXPERIMENTS

### 3.1 IMAGENET-1K CLASSIFICATION

**Dataset and Settings:** ImageNet-1K (Deng et al., 2009) is a classification benchmark with annotations of 1000 categories. It contains 1.2M training images and 50K validation images. In our evaluation, we report Top-1 accuracy (%) on a single-crop setting along with complexity metrics (measured in Parameters and FLOPs). We train our models with the standard settings: for 300 epochs with 224×224 resolution inputs. We use `timm` (Wightman, 2019) library for our implementation. We use the original hyperparameter settings for each of the backbone. We use increasing stochastic depth with ratio of 0.25, 0.4 for Swin-T+GrafT and Swin-S+GrafT respectively.

Table 1: Comparison of GrafT with pyramid architectures on ImageNet-1K.

| Model | Params (M) | FLOPs (G) | Acc. (%) | Model | Params (M) | FLOPs (G) | Acc. (%) |
|---|---|---|---|---|---|---|---|
| PVT-M (Wang et al., 2021) | 44 | 6.7 | 81.2 | PVT-L | 61 | 9.8 | 81.7 |
| PoolFormer-S36 (Yu et al., 2022) | 31 | 5.2 | 81.4 | PoolFormer-M36 | 56 | 9.1 | 82.1 |
| T2T$_t$-14 (Yuan et al., 2021) | 22 | 6.1 | 81.7 | T2T$_t$-19 | 39 | 9.8 | 82.2 |
| TNT-S (Han et al., 2021) | 24 | 5.2 | 81.5 | TNT-B | 66 | 14.1 | 82.9 |
| Swin-T (Liu et al., 2021) | 29 | 4.5 | 81.3 | Swin-S | 50 | 8.7 | 83 |
| ViL-S-RPB (Zhang et al., 2021) | 25 | 4.9 | 82.4 | ViL-M | 40 | 8.7 | 83.5 |
| RegionViT-S (Chen et al., 2022) | 31 | 5.3 | 82.6 | RegionViT-M | 41 | 7.4 | 83.1 |
| Swin-T+GrafT (ours) | 34 | 5.1 | (+1.4) 82.7 | Swin-S+GrafT | 64 | 9.7 | (+0.6) 83.6 |
| CSWin-T | 23 | 4.3 | 82.7 | - | - | - | - |
| CSWin-T+GrafT (ours) | 29 | 4.7 | (+0.5) 83.2 | - | - | - | - |

**Results on ImageNet-1K:** Table 1 and Table 2 shows the results of the GrafT applied on three well-known Transformers that have homogeneous structure and pyramid structure: DeiT, Swin, and CSWin. In the Table **??**, Swin-T+GrafT achieves 82.7% top-1 accuracy which is 1.4% better than Swin-T. Swin-S+GrafT achieves 83.6% Top-1 accu-

Table 2: Comparison of GrafT with homogeneous architectures on ImageNet-1K.

| Model | Params (M) | FLOPs (G) | Acc. (%) |
|---|---|---|---|
| DeiT-T (Touvron et al., 2021) | 5.7 | 1.3 | 72.2 |
| CrossViT-9 (Chen et al., 2021) | 8.6 | 1.8 | 73.9 |
| PVT-T (Wang et al., 2021) | 13.2 | 1.9 | 75.1 |
| DeiT-T + GrafT (ours) | 7.9 | 1.2 | (+3.9) 76.1 |

racy which is 0.6% better than Swin-S. The Figure 1-(b) shows that RegionViT outperforms Swin but Swin+GrafT outperforms RegionViT due to the support from GrafT. The gain by the GrafT is drawn by gray arrows. Secondly, CSWin-T+GrafT achieves 83.2% top-1 accuracy, a new SOTA result and it is 0.5% better than CSWin-T. Table 2 shows that DeiT-T+GrafT achieves 76.1% top-1 accuracy which is 3.9% better than DeiT-T. CrossViT outperforms DeiT but DeiT-T+GrafT further outperforms CrossViT due to the support from GrafT. In addition, even though DeiT-T+GrafT does not take advantage of the vertical pyramid structure, it outperforms PVT, a vision Transformer with the vertical pyramid structure. In DeiT-T+GrafT, the full attention in the backbone Transformer is replaced by the window-based local attention without window shifting to follow the suggested architecture in Figure 2-(b). We confirm that DeiT with the window-based local attention underperforms the one with the full attention by 2%. Therefore, GrafT delivers 5.9% performance gain (from 70.2% to 76.1%) by providing multi-scale high-level information and global receptive fields to the revised DeiT.

## 3.2 OBJECT DETECTION AND SEMANTIC SEGMENTATION

**Dataset and Settings for COCO2017:** COCO2017 (Lin et al., 2014) consists of 118K images. for training, 5K for validation and 20K for testing. We adopt Mask R-CNN(He et al., 2017) framework to evaluate Swin-T+GrafT which is pretrained on the ImageNet-1K dataset. We use the stochastic depth with ratio of 0.1 and 0.2 for the $1 \times (SS)$ schedule training and $3 \times$ (MS) schedule training and follow the original hyperparameter settings as Swin. Here, $1 \times$ (SS) means 12 training epochs with single scale, $3 \times (MS)$ means 36 training epochs with multi-scale.

Table 3: Performance of GrafT on object detection and instance segmentation on COCO2017.

| Model | Params | FLOPs | 1x (SS) | | 3x (MS) | |
|---|---|---|---|---|---|---|
| | | | AP$^b$ | AP$^m$ | AP$^b$ | AP$^m$ |
| ResNet50 (He et al., 2016) | 44 | 260 | 38.0 | 34.4 | 41.0 | 37.1 |
| PVT-S (Wang et al., 2021) | 44 | 245 | 40.4 | 37.8 | 43.0 | 39.9 |
| VIL-S-RPB (Zhang et al., 2021) | 45 | 277 | – | – | 47.1 | 42.7 |
| RegionViT-S (Chen et al., 2022) | 50 | 171 | 42.5 | 39.5 | 46.3 | 42.3 |
| Swin-T (Liu et al., 2021) | 48 | 264 | 42.2 | 39.1 | 46.0 | 41.6 |
| Swin-T+GrafT (ours) | 53 | 275 | (+1.1) 43.3 | (+0.8) 39.9 | (+1.0) 47.0 | (+0.9) 42.5 |
| VIL-S-RPB (Zhang et al., 2021) | 60 | 352 | – | – | 48.9 | 44.2 |
| RegionViT-B (Chen et al., 2022) | 92 | 288 | 43.5 | 40.1 | 47.2 | 43.0 |
| Swin-S (Liu et al., 2021) | 69 | 354 | 44.8 | 40.9 | 48.5 | 43.3 |
| Swin-S+GrafT (ours) | 84 | 373 | (+0.2) 45.0 | (+0.3) 41.2 | – | – |

**Results on Object Detection and Instance Segmentation:** Table 3 shows that Swin-T+GrafT outperforms other models in both object detection and instance segmentation while being trained with $1\times$ schedule and $3\times$ schedule. RegionViT-S outperforms Swin-T but Swin-T+GrafT fights back and outperform RegionViT-S again due to the support from GrafT. The multi-level semantics from GrafT plays an important role to improve both object detection and instance segmentation.

**Dataset and Settings for ADE20K:** ADE20K (Zhou et al., 2017) annotates 150 categories for semantic segmentation. It contains 20K training, 2K validation and 3K testing images. In our evaluations, we use multi-scale mIoU as the metric (using scales of [0.5, 0.75, 1.0, 1.25, 1.5, 1.75]$\times$ the training resolution) and follow a training procedure similar to Swin (Liu et al., 2021). We also report model complexity metrics such as parameters, FLOPs (for an input size of 512$\times$2048) and frame-rate. We use GrafT backbones pretrained on ImageNet-1K (Deng et al., 2009) for 300 epochs at a 224$\times$224 resolution, and finetune it with the decoder at a 512$\times$512 resolution. We choose UperNet (Xiao et al., 2018) as our decoder, and implement within the `mmsegmentation` (Contributors, 2020) framework. We use the original hyperparameter settings as Swin.

**Results on Semantic Segmentation:** Table 4 shows that Swin+GrafT outperform Swin in both single scale **(+1.0)** and multi-scale **(+1.3)** training settings. The multi-level semantics from GrafT is beneficial to segment various sizes of objects.

Table 4: Performance of GrafT on semantic segmentation on ADE20K.

| Models | Param (M) | FLOPs (G) | mIOU SS | mIOU MS |
|---|---|---|---|---|
| Swin-T (Liu et al., 2021) | 59 | 945 | 44.5 | 45.8 |
| Swin-T + Graft | 66 | 955 | (+1.0) 45.5 | (+1.3) 47.1 |

### 3.3 Ablations on ImageNet

We conducted the following experiments to better understand GrafT. Swin-T and Deit-T are used in the experiments.

Table 5: Different downsampling and upsampling approaches in GrafT. When ablating downsampling methods, we always consider Learnable W-Bilinear interpolation as upsampling, and when ablating upsampling methods, we use linear projection as downsampling.

| Downsampling | Params (M) | FLOPs (G) | Through. (im/s) | Acc. (%) | Upsampling | Params (M) | FLOPs (G) | Through. (im/s) | Acc. (%) |
|---|---|---|---|---|---|---|---|---|---|
| Linear projection | 8.5 | 1.3 | 2260 | 75.2 | Nearest neighbor | 8.4 | 1.3 | 2890 | 74.8 |
| Cross attention | 7.9 | 1.2 | 2834 | 75.4 | Cross attention | 8.7 | 1.3 | 2392 | 74.4 |
| Average pooling | 7.9 | 1.2 | 3143 | 76.1 | Learnable bilinear | 8.5 | 1.3 | 2260 | 75.2 |

**Downsampling & Upsampling in GrafT:.** Table 5 shows the performance of DeiT-T+GrafT on different horizontal downsampling and horizontal upsampling approaches. GrafT keeps learnable W-Bilinear interpolation as an upsampling method while applying various downsampling approaches. The linear projection approach concatenates spatially neighboring tokens into channels and applies linear projection to them to reduce the channels. The cross attention approach creates spatially coraser feature by average pooling the input feature. It then uses a coarser feature as a query and the input feature in the backbone transformer as key and value to perform cross attention. The average pooling approach creates a spatially coarser features by average pooling spatially neighboring tokens in the input feature. The result shows that the average pooling approach is the most efficient downsampling method in terms of memory and computation complexities and speed. GrafT keeps linear projection as a downsampling method while applying various upsampling approaches. Cross attention uses the fine feature in the backbone transformer as query and spatially coarser feature in GrafT as key and value to exchange global and local information. We also tried Nearest neighbor interpolation as the simplest upsampling method. Learnable W-Bilinear interpolation applies anti-aliasing weights to prevent the aliasing effect and applies bilinear interpolation in each local region separately. The result shows that learnable W-Bilinear interpolation achieves the highest accuracy with reasonable complexities and speed. Therefore, we adopt average pooling as a downsampling method and learnable W-Bilinear interpolation as an upsample method for GrafT.

Table 6: Considering different (a) number of scales within a GrafT, and (b) number of GrafTs in a model.

|              | (a)      |               |              |            |
|--------------|----------|---------------|--------------|------------|
| Model        | # scales | Params
(M) | FLOPs
(G) | Acc. (%)
(%) |
| Swin-T (Liu et al., 2021) | 0 | 29 | 4.5 | 81.3 |
| Swin-T + GrafT | 1 | 33.5 | 5.0 | (+1.0) 82.3 |
|                | 3 | 34.0 | 5.1 | (+1.4) 82.7 |

|              | (b)      |               |              |            |
|--------------|----------|---------------|--------------|------------|
| Model        | # GrafTs | Params
(M) | FLOPs
(G) | Acc. (%)
(%) |
| DeiT-T (Touvron et al., 2021) | 0 | 5.7 | 1.3 | 72.2 |
| DeiT-T + GrafT | 4 | 6.5 | 1.2 | (+2.2) 74.4 |
|                | 8 | 7.3 | 1.2 | (+3.4) 75.6 |
|                | 11 | 7.9 | 1.2 | (+3.9) 76.1 |

**The number of multi-scale features in GrafT:** GrafT exploits a horizontal pyramid structure where multi-scale low-resolution features are created. It is important to understand whether creating features having multiple high-level semantics is more beneficial than creating a single-scale feature. Table 6-(a) shows that exploiting features with three different scales achieves +1.4% better accuracy with a small computation increase than exploiting a single-scale feature. For example, at the first stage in Swin where the resolution of the input feature is $56 \times 56$, GrafT should create features with the size of $28 \times 28$, $14 \times 14$, $7 \times 7$ to deliver multi-scale global information.

**Performance over the number of GrafT:** Table 6-(b) shows the performance over the number of GrafTs in DeiT. The accuracy consistently increases as the number of GrafT increases. Therefore, we attach GrafT to all the vertical layers above the first layer in DeiT, Swin, and CSWin to achieve the best performance. GrafT is not attached to the first layer in the backbone Transformer because the feature needs to be encoded enough before being used to create coarse features in GrafT.

**Sharing FFN:** Table 7 compares the design of shared FFN vs. separate FFN between GrafT and the backbone transformer. DeiT is used as the backbone transformer in this experiment. In the first row, DeiT and GrafT have separate FFN, so the low-level feature in the DeiT and the high-level feature in GrafT are fused after FFN blocks. In the second row, DeiT and GrafT

Table 7: Sharing the parameters of backbone-FFN

| Shared FFN | Params (M) | FLOPs (G) | Acc. (%) |
|------------|------------|-----------|----------|
|            | 8.2 | 1.2 | 74.8 |
| ✓          | 7.9 | 1.2 | (+1.3) 76.1 |

have a shared FFN, so the low-level feature in the DeiT and the high-level feature in GrafT are fused before the shared FFN block. The design of shared FFN achieves +1.3% higher accuracy with fewer parameters than having a separate FFN. Therefore, we adopt the shared FFN in the GrafT architecture.

## 4 RELATED WORK

**Vision transformers:** Convolution neural networks (CNNs) have been widely adopted as it have shown promising performance (Krizhevsky et al., 2012; He et al., 2016; Chen et al., 2017; Howard et al., 2017; Sandler et al., 2018; Hu et al., 2018; Huang et al., 2017; Simonyan & Zisserman, 2014; Szegedy et al., 2015b; Tan & Le, 2019) on small-scale dataset such as ImageNet-1K (Deng et al., 2009). Inductive biases such as translation invariance and locality from CNNs are the key reasons to be trained well from scratch in small-scale dataset. Recently, Transformers (*e.g.,* ViT (Dosovitskiy et al., 2020) or DeiT (Touvron et al., 2021)) achieved comparable results to CNNs. First type is the transformer with homogeneous structure like ViT where the number of tokens and channels do not change over the vertical layers. T2T (Yuan et al., 2021) proposes progressive tokenization method where spatial structures are preserved. CrossViT (Chen et al., 2021) creates two branches to formulate both local and global information and exchange information. PiT (Heo et al., 2021) and PVT (Wang et al., 2021) successfully applied pyramid structure into transformer by dividing the vertical layers into multiple stages and progressively decrease the number of tokens and increase the channels over stages. Swin (Liu et al., 2021) introduces window self-attention where self-attention is performed in each window and shifting window mechanism exchanges information among windows with pyramid structure. RegionViT (Chen et al., 2022) creates two branches to formulate local tokens and global tokens like CrossViT and assign each global token to local tokens in the same region to exchange information. CSWin (Dong et al., 2022) introduces cross-shaped window self-attention where half of the channels is used to create vertical stripes as local regions and the other half is used to create horizontal stripes as local regions. In this paper, we propose GrafT, a simple

and cost-efficient add-on that provides rich global information to backbone transformers where there is a lack of communication between local regions because of local self-attention.

Table 8: Approaches to deliver high-level semantics in terms of their structure or multiple scales.

| Model | Vertical structure | # MS per layer | Fusion method per layer |
|---|---|---|---|
| ViT-T (DeiT-T) (Touvron et al., 2021) | Homogeneous | None | None |
| CrossViT-9 (Chen et al., 2021) | Homogeneous | 2 | Cross attention |
| DeiT-T + GrafT | Homogeneous | 2 | Learnable W-Bilinear + E-Wise add. |
| PVT-T (Wang et al., 2021) | Pyramid | None | None |
| T2T$_t$(Yuan et al., 2021) | Homogeneous | None | None |
| PoolFormer-S12 (Yu et al., 2022) | Pyramid | None | None |
| TNT-S (Han et al., 2021) | Homogeneous | 2 | LL + E-Wise add. |
| Swin-T (Liu et al., 2021) | Pyramid | None | None |
| RegionViT-S (Chen et al., 2022) | Pyramid | 2 | Cross attention |
| Swin-T + GrafT | Pyramid | 4 | Learnable W-Bilinear + E-Wise add. |

**Exploiting multi-scale global tokens:** While pyramid structure transformers (*e.g.,* Swin) learns multi-scale features in a hierachical way, high-level semantics is introduced at the last layers and transformer cannot receive early guidance on how to efficiently encode low-level semantics by understanding the high-level semantics. Some transformers such as CrossViT (Chen et al., 2021), TNT (Han et al., 2021), RegionViT (Chen et al., 2022) keep two branches to encode low-level semantics and high-level semantics from the early stage. However, having two separate branches are detrimental to throughput and requires careful design of choosing which layers to exchange local and global information and choosing the right size ratio of low-resolution and high-resolution features as mentioned in CrossViT (Chen et al., 2021) or the divisibility between local and global tokens. ViL (Zhang et al., 2021) creates global tokens through random initialization in each layer and use them to exchange information between local regions. However, this global token does not contain good inductive bias of local tokens and does not incorporate multiple high-level semantics. Table 8 summarizes the difference between GrafT and previous works on how to deliver high-level semantics. GrafT is unique in the sense that it delivers multiple high-level semantics by exploiting horizontal pyramid structure and uses simple element-wise addition to fuse global information to the backbone transformer. It is applicable to transformers with both homogeneous structure and pyramid structure and improves the performance of transformers without increasing the computation complexity due to the light-weight components as described in 2.2.

## 5 CONCLUSION

In this paper, we introduced GrafT: an add-on component which can easily be adopted in both homogeneous and pyramid vision transformers, enabling multi-scale feature fusion in arbitrary depths of a model. The proposed GrafT branches are designed to be efficient, relying on the backbone to perform heavy computations. In fact, it gives consistent gains at a minimal computation burden. We also validated its effectiveness across multiple backbones and benchmarks including classification, detection and segmentation. In the current work, GrafT is applied to three well-known Transformers: DeiT, Swin, and CSWin. Going forward, we hope that GrafT becomes a generally used component for introducing multi-scale features, which particularly benefits tasks with high-resolution inputs.

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

# A APPENDIX

Table A: Grafting towards left (upscale from backbone) or towards right (downscale from backbobe).

| Model | left levels | right levels | Params (M) | FLOPs (G) | Top-1 (%) |
|---|---|---|---|---|---|
| DeiT-T (Touvron et al., 2021) | 0 | 0 | 5.7 | 1.3 | 72.2 |
| DeiT-T + GrafT | 1 | 1 | 8.0 | 1.6 | 75.1 |
| | 0 | 2 | 10.0 | 1.3 | 75.6 |
| | 0 | 1 | 7.3 | 1.2 | 75.6 |

**GrafT should be used to generate higher-level semantics rather than lower-level semantics:** Table A shows the performance over graphting directions. In the experiment, downsampling uses the average pooling and upsampling uses learnable W-Bilinear interpolation for both left and right grapftings. In DeiT-T + GrafT, 1 level of right grafting per layer provides the best accuracy with the least memory and computation complexity. Left grafting is not helpful because the hallucinated higher-resolution feature map in the GrafT may not contain additional local information. It is not helpful to perform 2 levels of right grafting in DeiT because 2nd level of Graft has a feature map with the size of $2 \times 2$ and it is too low-resolution feature map that it does not provide any additional global information.

