# OpenReview forum: "Grafting Vision Transformers"
_ICLR.cc/2023/Conference — Submitted to ICLR 2023_

### Official Review · Reviewer_wB1L · 2022-10-25

**Confidence:** 5
**Correctness:** 3
**Technical Novelty And Significance:** 3
**Empirical Novelty And Significance:** Not applicable
**Recommendation:** 6

**Clarity, Quality, Novelty And Reproducibility:**

The paper has good clarity and the proposed method is novel and validated with ablation studies. Reproducibility is medium as the method involves lots of implementation details.

**Strength And Weaknesses:**

Strength:
		The idea of adding such multi-scale attention branch in parallel is interesting and design choices of downsampling and upsampling are carefully validated empirically.

	Weaknesses:
1.	The paper says the method adds minimal complexity. The proposed module actually introduce quite the parameters and flops, 17% #param increase and 14% FLOPs increase when applied to Swin-T, 29% #param increase and 11% FLOPs increase for Swin-S.  And the model performance, compared with RegionViT-S, has only marginal 0.1 improvement, but 10% more #params.
2.	Can authors show the performance when applied to large models, e.g. Swin-B?
3.	Throughputs and memory? As the method adds parallel branch to existing architectures with consecutive downsamling and upsampling, which is hard for parallelization,  can authors show some throughputs comparison w v.s w/o GrafT. Also, each layer will have much more token maps, can authors compare the memory consumption as well?


**Summary Of The Paper:**

	The paper proposes to add a FPN-like parallel branch to existing vision transformers that fuses multi-scale tokens with consecutive downsampling, upsampling and window attention. Design choices of each part are studies empirically and improvement over Swin-T on ImageNet, COCO, ADE20k are reported.

**Summary Of The Review:**

Overall, the paper proposes a novel way to fuse multi-scale information in vision transformers, which is to add a FPN-like parallel branch to each of the attention block. It would make the paper better if some concerns regarding computation, throughputs, memory and applying to large models are resolved.

---

> ### Author Response · Authors · 2022-11-19
> **Response to Reviewer wB1L - (1)**
>
>
> - **Question**: The paper says the method adds minimal complexity. The proposed module actually introduce quite the parameters and flops, 17% #param increase and 14% FLOPs increase when applied to Swin-T, 29% #param increase and 11% FLOPs increase for Swin-S.  And the model performance, compared with RegionViT-S, has only marginal 0.1 improvement, but 10% more #params.
>
> **Answer:** Thank you for the detailed comment. As shown in Table c1, we applied GrafT on CSWin-T to show the performance of GrafT on the stronger backbone. CSWin-T+GrafT achieves a new SOTA 83.2% accuracy which is +0.6% better than RegionViT-S while exploiting less compute and parameters. It increases 9% Flops and 32% parameters of CSWin-T. GrafT is designed to add minimal compute to obtain higher performance whlie sacrificing relatively more parameters. However, as shown in Table c1 and c2, GrafT adds only about 1% memory to Swin-T, Swin-S and 3% memory to CSWin-T. The increase in parameters merely increase memory size. Therefore, GrafT is still suitable for Transformers performing mobile vision tasks that require low latency and low memory.
>
> **[Table c1]**. GrafT on models in 20-35M size for ImageNet-1K
> | Model                    | Params (M) | FLOPs (G) |     Acc(%)      | Thrput (imgs/sec) | Mem (MiB) |
> | ------------------------ |:----------:|:---------:|:---------------:|:-----------------:|:---------:|
> | ViL-S-RPB                |     25     |    4.9    |      82.4       |        --         |    --     |
> | RegionViT-S              |     31     |    5.3    |      82.6       |        988        |   4633    |
> | Swin-T                   |     29     |    4.5    |      81.3       |       1387        |   4257    |
> | Swin-T+GrafT **(ours)**  |     34     |    5.1    |   (+1.4) 82.7   |       1007        |   4307    |
> | CSWin-T                  |     23     |    4.3    |      82.7       |       1179        |   3195    |
> | CSWin-T+GrafT **(ours)** |     29     |    4.7    | **(+0.5) 83.2** |        949        |   3289    |
>
> **[Table c2]** GrafT on models in 35-70M size on ImageNet-1K
>
> | Model                    | Params (M) | FLOPs (G) |   Acc(%)    | Thrput (imgs/sec) | Mem (MiB) |
> | ------------------------ |:----------:|:---------:|:-----------:|:-----------------:|:---------:|
> | ViL-M-RPB                |     40     |    8.7    |    83.5     |        --         |    --     |
> | RegionViT-M              |     41     |    7.4    |    83.1     |        825        |   4481    |
> | Swin-S                   |     50     |    8.7    |     83      |        859        |   4357    |
> | Swin-S+GrafT **(ours)**  |     64     |    9.7    | (+0.6) 83.6 |        646        |   4405    |

---

> > ### Author Response · Authors · 2022-11-19
> > **Response to Reviewer wB1L - (2)**
> >
> >
> > - **Question**: Can authors show the performance when applied to large models, e.g. Swin-B?
> >
> >
> > **Answer:**  Thank you for interesting question. In Table c3, we applied GrafT on Swin-B to show its applicability on big models in 70-120M size. Swin-B+GrafT achieves (+0.2%)83.7% accuracy. It is a decent improvement considering the Swin-B already has a high accuracy level 83.5%. However, we would like to emphasize that GrafT is designed to impose a minimal amount of compute on the backbone Transformer to improve the performance on various downstream tasks. Therefore, GrafT is more suitable for Transformers in super tiny(1-10M), tiny(20-35M), and small (35-70M) sizes which are desirable for mobile vision tasks that require resource-constrained devices.
> >
> > For the reviewer's convenience, we added Table c4, where GrafT is applied on MobileViT, a super tiny model in 1-10M size. We run the model on ImageNet100 for 100 epochs while following the hyperparameters from the MobileViT. The results are the following.
> >
> > *  MobileViT_xxs+GrafT achieves (+1.3%)73.5% accuracy by using only +6% more flop and 14% more parameters than MobileViT_xxs.
> > *  MobileViT_xs+GrafT achieves (+2.3%)76.9% accuracy by using only +5% more flop and 16% more parameters than MobileViT_xs. Interestingly enough, MobileViT_xs+GrafT outperforms MobileViT_s by +0.5% while using only 54% compute, 48% parameters, and having a similar throughput of the MobileViT_s.
> > *  MobileViT_s+GrafT achieves (+1.6%)78.0% accuracy by using only +5% more flop and 15% more parameters than MobileViT_s.
> >
> > **[Table c3]** GrafT on models in 70-120M size on ImageNet-1K
> > | Model                   | Params (M) | FLOPs (G) |   Acc(%)    | Thrput (imgs/sec) | Mem (MiB) |
> > | ----------------------- |:----------:|:---------:|:-----------:|:-----------------:|:---------:|
> > | Swin-B                  |     88     |   15.4    |    83.5     |        638        |   5225    |
> > | Swin-B+GrafT **(ours)** |    114     |   17.1    | (+0.2) 83.7 |        469        |   5245    |
> >
> >
> > **[Table c4]** GrafT on models in 1-10M for ImageNet100
> >
> > | Model                          | Params (M) | FLOPs (G) |   Acc(%)   | Thrput (imgs/sec) |
> > | ------------------------------ |:----------:|:---------:|:----------:|:-----------------:|
> > | MobileVit_xxs                  |    1.27    |   0.42    |    72.2    |       3784        |
> > | MobileVit_xxs+GrafT **(ours)** |    1.45    |   0.44    | (+1.3)73.5 |       2576        |
> > | MobileVit_xs                   |    2.32    |   1.05    |    74.6    |       2127        |
> > | MobileVit_xs+GrafT **(ours)**  |    2.69    |   1.10    | (+2.3)76.9 |       1656        |
> > | MobileVit_s                    |    5.58    |   2.03    |    76.4    |       1662        |
> > | MobileVit_s+GrafT **(ours)**   |    6.44    |   2.13    | (+1.6)78.0 |       1334        |
> >
> > - **Question**: Throughputs and memory? As the method adds parallel branch to existing architectures with consecutive downsamling and upsampling, which is hard for parallelization, can authors show some throughputs comparison w v.s w/o GrafT. Also, each layer will have much more token maps, can authors compare the memory consumption as well?
> >
> > **Answer:** Thank you for your insightful question. We recorded inference throughput and memory in Table c1, c2, and c3. GrafT has 81%, 73%, 75%, 74% throughput on CSWin-T, Swin-T, Swin-S, Swin-B. It increases 3%, 1%, 1%, 1% memory on CSWin-T, Swin-T, Swin-S, Swin-B. The throughput and memory are measured in NVIDIA RTX A5000 by having 128 batch sizes with one GPU for all models.

---

### Official Review · Reviewer_EcB8 · 2022-10-25

**Confidence:** 5
**Correctness:** 3
**Technical Novelty And Significance:** 3
**Empirical Novelty And Significance:** Not applicable
**Recommendation:** 3

**Clarity, Quality, Novelty And Reproducibility:**

The experiments are insufficient to demonstrate the effectiveness of the proposed method.

**Strength And Weaknesses:**

Strength:

The proposed method is evaluated on several fundamental vision tasks, although the evaluation on each task is limited.


Weaknesses:

The performance of ViL (Zhang et al., 2021) is much lower than that in the original paper. Please check the experiments and use ViL as the baseline to evaluate the improvement because ViL is the state-of-the-art method.

This paper should add the proposed GrafT to models with different model sizes. In the current version, the proposed method is only evaluated for Swin-T for object detection, instance segmentation and semantic segmentation. It is only evaluated for Swin-T, Swin-S and DeiT-T for image classification, which is insufficient to demonstrate the effectiveness of the proposed method.


**Summary Of The Paper:**

This paper presents an add-on component (termed GrafT) that considers global dependencies and multi-scale information throughout the transformer network. The proposed method is evaluated on several fundamental visual recognition problems.

**Summary Of The Review:**

The insufficient experiments and unfair comparisons are my main concerns. Please see the above comments.

---

> ### Author Response · Authors · 2022-11-19
> **Response to Reviewer EcB8 - (1)**
>
> - **Question:** The performance of ViL (Zhang et al., 2021) is much lower than that in the original paper. Please check the experiments and use ViL as the baseline to evaluate the improvement because ViL is the state-of-the-art method.
>
> **Answer:** We appolgize and we updated the paper (Table1 and Table 3). We got the old numbers from RegionVit paper, which used the numbers of the ViL's 1st version paper in the arxiv. ViL has multiple different numbers reported in the papers, its gihub page, and the RegionViT paper. We corrected the performance of ViL in our updated paper by using the ViL's 2nd version paper in the arxiv and colored them in red. Thank you for your correction.
>
> Following the suggestion of the reviewer, we applied GrafT on top of a stronger SOTA model, CSWin. Table b2 shows that CSWin-T+GrafT achieves (+0.5%)83.2% on ImageNet which is the new state-of-the-art among models with 25-35M parameters. We plan to experiment ViL+GrafT in the final version of the paper.
>
>
> - **Question:** This paper should add the proposed GrafT to models with different model sizes. In the current version, the proposed method is only evaluated for Swin-T for object detection, instance segmentation and semantic segmentation. It is only evaluated for Swin-T, Swin-S and DeiT-T for image classification, which is insufficient to demonstrate the effectiveness of the proposed method.
>
> **Answer:** Following the suggestion, we tested additional models with different sizes. We tested GrafT on top of MobileViT, CSWin-T, Swin-B on classifications and measured the performance of Swin-S+GrafT on detection and instance segmentation tasks.
>
> In Table b1, we applied GrafT on MobileViT to show its applicability on super tiny models in 1-10M size. We run the model on ImageNet100 for 100 epochs while following the hyperparameters from the MobileViT.
> *  MobileViT_xxs+GrafT achieves (+1.3%)73.5% accuracy by using only +6% more flop and 14% more parameters than MobileViT_xxs.
> *  MobileViT_xs+GrafT achieves (+2.3%)76.9% accuracy by using only +5% more flop and 16% more parameters than MobileViT_xs. Interestingly enough, MobileViT_xs+GrafT outperforms MobileViT_s by +0.5% while using only 54% compute, 48% parameters, and having a similar throughput of the MobileViT_s.
> *  MobileViT_s+GrafT achieves (+1.6%)78.0% accuracy by using only +5% more flop and 15% more parameters than MobileViT_s.
>
>
> **[Table b1]** GrafT on models in 1-10M for ImageNet100
>
> | Model                          | Params (M) | FLOPs (G) |   Acc(%)   | Thrput (imgs/sec) |
> | ------------------------------ |:----------:|:---------:|:----------:|:-----------------:|
> | MobileVit_xxs                  |    1.27    |   0.42    |    72.2    |       3784        |
> | MobileVit_xxs+GrafT **(ours)** |    1.45    |   0.44    | (+1.3)73.5 |       2576        |
> | MobileVit_xs                   |    2.32    |   1.05    |    74.6    |       2127        |
> | MobileVit_xs+GrafT **(ours)**  |    2.69    |   1.10    | (+2.3)76.9 |       1656        |
> | MobileVit_s                    |    5.58    |   2.03    |    76.4    |       1662        |
> | MobileVit_s+GrafT **(ours)**   |    6.44    |   2.13    | (+1.6)78.0 |       1334        |
>
> In Table b2, we applied GrafT on CSWin-T to show its applicability on tiny models in 25-35M parameters. CSWin-T+GrafT achieves a new SOTA 83.2% accuracy which is +0.8% and +0.6% better than ViL-S-RPB and RegionViT-S, respectively, while exploiting similar compute and parameters. GrafT on Swin-T and CSWin-T shows that it can help Transformers with 25-35M size achieve higher accuracy.
>
>
> **[Table b2]** GrafT on models in 20-35M size for ImageNet-1K
> | Model                    | Params (M) | FLOPs (G) |     Acc(%)      | Thrput (imgs/sec) | Mem (MiB) |
> | ------------------------ |:----------:|:---------:|:---------------:|:-----------------:|:---------:|
> | ViL-S-RPB                |     25     |    4.9    |      82.4       |        --         |    --     |
> | RegionViT-S              |     31     |    5.3    |      82.6       |        988        |   4633    |
> | Swin-T                   |     29     |    4.5    |      81.3       |       1387        |   4257    |
> | Swin-T+GrafT **(ours)**  |     34     |    5.1    |   (+1.4) 82.7   |       1007        |   4307    |
> | CSWin-T                  |     23     |    4.3    |      82.7       |       1179        |   3195    |
> | CSWin-T+GrafT **(ours)** |     29     |    4.7    | **(+0.5) 83.2** |        949        |   3289    |

---

> > ### Author Response · Authors · 2022-11-19
> > **Response to Reviewer EcB8 - (2)**
> >
> > We continue to answer the question above.
> >
> > In Table b3, we applied GrafT on Swin-B to show its applicability on big models in 70-120M size. Swin-B+GrafT achieves (+0.2%)83.7% accuracy. It is a decent improvement considering the Swin-B already has a high accuracy level 83.5%. However, we would like to emphasize that GrafT is designed to impose a minimal amount of compute on the backbone Transformer to improve the performance on various downstream tasks. Therefore, GrafT is more suitable for Transformers in super tiny(1-10M), tiny(20-35M), and small (35-70M) sizes which are desirable for mobile vision tasks that require resource-constrained devices.
> >
> > **[Table b3]** GrafT on models in 70-120M size on ImageNet-1K
> > | Model                   | Params (M) | FLOPs (G) |   Acc(%)    | Thrput (imgs/sec) | Mem (MiB) |
> > | ----------------------- |:----------:|:---------:|:-----------:|:-----------------:|:---------:|
> > | Swin-B                  |     88     |   15.4    |    83.5     |        638        |   5225    |
> > | Swin-B+GrafT **(ours)** |    114     |   17.1    | (+0.2) 83.7 |        469        |   5245    |
> >
> > In Table b4, we measure the performance of Swin-S+GrafT on object detection and instance segmentation in 1× schedule (12 epochs) setting. Swin-S+GrafT achieves (+0.2%)45.0% mAP and (+0.3%)41.2% mAP on the detection and instance segmentation, respectively. It shows that GrafT is beneficial for Transformers with 35-70M size to perform better on various downstream tasks, including classification, detection, and instance segmentation.
> >
> >
> > **[Table b4]** Swin-S+GrafT on Detection & Instance Segmentation
> > | Model                   | Params (M) | FLOPs (G) | AP(b) 1x   | AP(m) 1x   |
> > | ----------------------- | ---------- | --------- | ---------- | ---------- |
> > | RegionViT-B             | 83         | 309       | 43.5       | 40.1       |
> > | Swin-S                  | 69         | 354       | 44.8       | 40.9       |
> > | Swin-S+GrafT **(ours)** | 84         | 373       | (+0.2)45.0 | (+0.3)41.2 |
> >
> > Due to the compute and time limit during the rebuttal, we were able to conduct experiments on four more model sizes(MobileViT_xxs, MobileViT_xs, MobileViT_s, Swin-B), two more models(CSWin, MobileViT), and Detection and instance segmentation on Swin-S+GrafT. We will add experiments on ViL+GrafT and more segmentation tasks in our final version.

---

### Official Review · Reviewer_br4Z · 2022-10-25

**Confidence:** 3
**Correctness:** 4
**Technical Novelty And Significance:** 2
**Empirical Novelty And Significance:** 2
**Recommendation:** 6

**Clarity, Quality, Novelty And Reproducibility:**

There are some details in experiments that should be clarified, as described in the weakness.

**Strength And Weaknesses:**

Strengths
* The description of GrafT is clear and the figures are nice, making the paper easy to understand.
* The authors empirically demonstrate the effectiveness of GrafT on multiple tasks.

Weakness
* The improvement on ImageNet-1K is marginal. In Table 1, Swin-T+ GrafT outperforms RegionViT-S by only 0.1 but introduces 3.4M more parameters.
* GrafT should generally introduce more FLOPs and parameters for the backbone model.  However, in Table 2, the FLOPs of DeiT-T+GrafT are even smaller than the original DeiT-T.  Moreover, are the training settings for DeiT+ GrafT and DeiT-T fair?
* The ablation in Table 6-(a) is not sufficient. What are the results with more scales? Is scale = 3 the best?
* Generalization performance. The authors only validate the effectiveness of GrafT on Transformer-based architectures. If the multi-scale features are beneficial, I wonder about the performance of GrafT combined with classical CNN models such as VGG and ResNet.

**Summary Of The Paper:**

This paper proposes an efficient and general component GrafT for Transformer-based vision architectures. The GrafT can introduce multi-scale features for early-stage layers, thus achieving better results. The authors demonstrate the effectiveness of the proposed GrafT on several tasks and datasets.


**Summary Of The Review:**

I have several concerns claimed in the weakness part, thus I choose marginally below the acceptance.
I may modify the score according to the author's response.

---

> ### Author Response · Authors · 2022-11-19
> **Response to Reviewer br4Z - (1)**
>
> - **Question:** The improvement on ImageNet-1K is marginal. In Table 1, Swin-T+ GrafT outperforms RegionViT-S by only 0.1 but introduces 3.4M more parameters.
>
> **Answer:** Thank you for detailed comment. As shown in Table a1, we applied GrafT on CSWin-T to show the performance of GrafT on the stronger backbone. CSWin-T+GrafT achieves a new SOTA 83.2% accuracy which is +0.6% better than RegionViT-S while exploiting less compute and parameters.
>
> **[Table a1]**. GrafT on models in 20-35M size for ImageNet-1K
> | Model                    | Params (M) | FLOPs (G) |     Acc(%)      | Thrput (imgs/sec) | Mem (MiB) |
> | ------------------------ |:----------:|:---------:|:---------------:|:-----------------:|:---------:|
> | ViL-S-RPB                |     25     |    4.9    |      82.4       |        --         |    --     |
> | RegionViT-S              |     31     |    5.3    |      82.6       |        988        |   4633    |
> | Swin-T                   |     29     |    4.5    |      81.3       |       1387        |   4257    |
> | Swin-T+GrafT **(ours)**  |     34     |    5.1    |   (+1.4) 82.7   |       1007        |   4307    |
> | CSWin-T                  |     23     |    4.3    |      82.7       |       1179        |   3195    |
> | CSWin-T+GrafT **(ours)** |     29     |    4.7    | **(+0.5) 83.2** |        949        |   3289    |
>
> - **Question:** GrafT should generally introduce more FLOPs and parameters for the backbone model. However, in Table 2, the FLOPs of DeiT-T+GrafT are even smaller than the original DeiT-T. Moreover, are the training settings for DeiT+ GrafT and DeiT-T fair?
>
> **Answer:** Thank you for your insightful question. In Figure2-(b), we suggested that the backbone and GrafT use the same local-attention method. To follow this design, we replaced the full-attention of DeiT with window-based local-attention **without window shifting**. Since the receptive field of the revised DeiT backbone is limited within each window region, it **underperforms** the original DeiT that employs full-attention. We confirmed that the DeiT with the window-based local-attention **without window shifting** achives **only 70.2%**, which is 2% lower than the original DeiT with full-attention. Therefore, **GrafT delivers 5.9% performance gain (from 70.2% to 76.1%)** by providing multi-scale high-level information and global receptive fields to the revised DeiT. Flops are reduced as the revised DeiT exploits window-based local attention that requires less compute in self-attention than full-attention. The training hyperparameters for the DeiT+GrafT are identical to the original DeiT.
>
> - **Question:** The ablation in Table 6-(a) is not sufficient. What are the results with more scales? Is scale = 3 the best?
>
> **Answer:** Thank you for your professional comment. As pyramid structures in well-known Transformers, such as PVT, Swin, and CSWin, use three levels of downsampling in vertical layers, GrafT also performs three levels of downsampling in horizontal layers. We believe that many Transformers with pyramid structures utilize only three levels of downsampling because the fourth downsampled feature map has a too low resolution, which does not deliver additional global information on top of the third downsampled feature map. For example, if the input size is (224x224), the size of the fourth downsampled feature map is (3x3). It is quite low resolution to provide additional information when (7x7) downsampled feature map is encoded. In addition, we confirmed that three levels of downsampling are more effective than one level of downsampling, as shown in Table 6-(a).

---

> > ### Author Response · Authors · 2022-11-19
> > **Response to Reviewer br4Z - (2)**
> >
> >
> > - **Question:** Generalization performance. The authors only validate the effectiveness of GrafT on Transformer-based architectures. If the multi-scale features are beneficial, I wonder about the performance of GrafT combined with classical CNN models such as VGG and ResNet.
> >
> > **Answer:** Thank you for providing the interesting suggestion. As shown in Figure 2-(b), the current design of GrafT requires the backbone model to have local-attention + FFN architecture so that GrafT can perform separate local-attention and share FFN. Since CNN-based models does not have self-attention + FFN structures, they are not proper candidates for GrafT. However, we can try to measure the performance of GrafT on CNN by deploying a hybrid model that use both CNN and Transformer. As shown in Table a2, we applied GrafT on MobileViT, a SOTA hybrid model, and obtained consistent gains over various model sizes. We ran the model on ImageNet100 for 100 epochs while following the hyperparameters from the MobileViT.
> >
> > *  MobileViT_xxs+GrafT achieves (+1.3%)73.5% accuracy by using only +6% more flop and 14% more parameters than MobileViT_xxs.
> > *  MobileViT_xs+GrafT achieves (+2.3%)76.9% accuracy by using only +5% more flop and 16% more parameters than MobileViT_xs. Interestingly enough, MobileViT_xs+GrafT outperforms MobileViT_s by +0.5% while using only 54% compute, 48% parameters, and having a similar throughput of the MobileViT_s.
> > *  MobileViT_s+GrafT achieves (+1.6%)78.0% accuracy by using only +5% more flop and 15% more parameters than MobileViT_s.
> >
> >
> > **[Table a2]** GrafT on models in 1-10M for ImageNet100
> >
> > | Model                          | Params (M) | FLOPs (G) |   Acc(%)   | Thrput (imgs/sec) |
> > | ------------------------------ |:----------:|:---------:|:----------:|:-----------------:|
> > | MobileVit_xxs                  |    1.27    |   0.42    |    72.2    |       3784        |
> > | MobileVit_xxs+GrafT **(ours)** |    1.45    |   0.44    | (+1.3)73.5 |       2576        |
> > | MobileVit_xs                   |    2.32    |   1.05    |    74.6    |       2127        |
> > | MobileVit_xs+GrafT **(ours)**  |    2.69    |   1.10    | (+2.3)76.9 |       1656        |
> > | MobileVit_s                    |    5.58    |   2.03    |    76.4    |       1662        |
> > | MobileVit_s+GrafT **(ours)**   |    6.44    |   2.13    | (+1.6)78.0 |       1334        |

---

### Author Response · Authors · 2022-11-19
**Summary**

We truly appreciate for reviewers' helpful feedback. In summary, we report following additional expereiments.
* Classification with two more models (CSWin, MobileViT)
* Classification with four more model sizes (MobileViT_xxs, MobileViT_xs, MobileViT_s, Swin-B),
* Detection and instance segmentation on Swin-S+GrafT.

The proposed GrafT can increase the performance of the existing Transformers with **minimal computational burdens**. It means we focus on **light-weight and low-latency Transformers, whose size ranges from 1-70M**. They are important for resource-constrained devices to perform mobile vision tasks. In our experiment, MobileViT, DeiT-T, CSWin-T, Swin-T, and Swin-S have model sizes that are most suitable for GrafT.

Addditionaly, we conducted new experiment CSWin+GrafT. This is the new SOTA in the 20-35M parameters range models. This outperforms CSWin-T, Swin-T, ViL-S-RPB, RegionViT-S by +0.5%, 1.9%, 0.8%, 0.6% respectively while manitaining similar flops. We added a corresponding table in each response to reviewers.

---

> ### Author Response · Authors · 2022-12-02
> **Summary - 2**
>
> Please allow us to add the reference of the MobileViT [1] and CSWin [2], two new models on which we testest GrafT during the rebuttal period. We will add the experimental results obtained during the rebuttal period in the final version of the paper.
>
> [1] Sachin Mehta and Mohammad Rastegari. Mobilevit: light-weight, general-purpose, and mobile-friendly vision transformer. arXiv preprint arXiv:2110.02178, 2021.
>
> [2] Xiaoyi Dong, Jianmin Bao, Dongdong Chen, Weiming Zhang, Nenghai Yu, Lu Yuan, Dong Chen, and Baining Guo. Cswin transformer: A general vision transformer backbone with cross-shaped windows. In Proceedings of the IEEE/CVF Conference on Computer Vision and Pattern Recognition (CVPR), pp. 12124–12134, June 2022.

---

### Decision · Program_Chairs · 2023-01-20

**Decision:**

Reject

**Justification For Why Not Higher Score:**

Even though the rebuttal added and clarified several aspects, the evaluation regarding model size is only evaluated on one specific architecture. Further the newly reported number of paramaters indicated that GrafT indeed increases the sumber of parameters to achieve better results. An exact ablation with comparable model sizes would be much stronger here.

**Justification For Why Not Lower Score:**

N/A

**Metareview: Summary, Strengths And Weaknesses:**

This paper proposes an efficient  add-on for Transformer-based vision architectures, calles GrafT. GrafT allows to introduce multi-scale features for early-stage layers in the network and can thereby lead to improved results.
The proposed results improve over the presented baseline , yet there have been concerns on the correctness of reported resultsin the first submission, as well as for example  the number of flops and the dependence on the model sizes.
The rebuttal addressed some of these concerns, yet could not fully resolve them in the current form. We strongly encourage the authors to pepare a revised version of the paper including the additional experiments as well as the corrected number to prepare a submission for  future submissions.



**Summary Of Ac-Reviewer Meeting:**

N/A